# Synthesis of Celecoxib-Eutectic Mixture Particles via Supercritical CO_2_ Process and Celecoxib Immediate Release Tablet Formulation by Quality by Design Approach

**DOI:** 10.3390/pharmaceutics14081549

**Published:** 2022-07-26

**Authors:** Seung-Hyeon Hong, Linh Dinh, Sharif Md Abuzar, Eun Seok Lee, Sung-Joo Hwang

**Affiliations:** 1College of Pharmacy, Yonsei University, 85 Songdogwahak-ro, Yeonsu-gu, Incheon 21983, Korea; shhongmartin@naver.com (S.-H.H.); dinhkhanhlinh@yonsei.ac.kr (L.D.); sumonzar@gmail.com (S.M.A.); leslech@naver.com (E.S.L.); 2Yonsei Institute of Pharmaceutical Sciences, Yonsei University, 85 Songdogwahak-ro, Yeonsu-gu, Incheon 21983, Korea

**Keywords:** celecoxib, supercritical fluid, eutectic mixture, adipic acid, saccharin, spray drying, quality by design

## Abstract

Significant improvements in the wettability and dissolution rate of celecoxib (CEL), a poorly soluble selective cyclooxygenase-2 (COX-2) inhibitor, have been shown by Huyn et al., 2019 by combining the binary pharmaceutical compositions including CEL and one of the two co-formers, adipic acid (ADI) and saccharin (SAC), into eutectic mixtures (EM). *Purpose:* In this study, we developed a therapeutic eutectic system for CEL which is a promising approach for oral delivery to enhance bioavailability. CEL EM were synthesized by novel techniques including supercritical CO_2_ techniques and new tablet formulations were purposed. *Methods:* CEL EM were synthesized by evaporation crystallization method, spray drying, supercritical fluid (SCF) techniques. The CEL EM particles were then characterized by differential scanning calorimetry, powder X-ray diffraction, Fourier-transform infrared spectroscopy, scanning electron microscope, and particle size analysis. Dissolution studies were carried out. With a quality by design approach, a statistical method through design of experiment and data analysis by JMP^®^ (SAS institute) was applied to CEL EM immediate release tablet formulation development. *Results:* CEL EM produced by spray drying technique, supercritical fluid (SCF) techniques were identified and characterized. The enhancement of dissolution was observed for SCF processed samples. The design space for CEL-ADI EM IR tablet and control limits for individual parameters were determined.

## 1. Introduction

Non-steroidal anti-inflammatory drugs (NSAIDs) are the most commonly used drug class for reducing pain, decreasing fever, preventing blood clots and, in higher doses, showing anti-inflammation effects. NSAIDs inhibit the cyclooxygenase (COX) enzyme family, which catalyzes the metabolism of arachidonic acid to prostaglandins, prostacyclin, and thromboxane [1]. COX has two isoforms (COX-1 and COX-2) and inhibition of COX-1 causes gastro intestinal side effects so the introduction of the first selective COX-2 inhibitor (375-fold selectivity) [2] in the pharmaceutical market revolutionized choice among NSAIDs. Celecoxib (CEL), 4-[5-(4-methylphenyl)-3-(trifluoromethyl)-1Hpyrazol-1-yl] benzene sulphonamide is an NSAID that selectively inhibit cyclooxygenase-2 (COX-2) enzymes [1,2]. Compared to other NSAIDs (e.g., naproxen, diclofenac), CEL has shown better efficacy in osteoarthritis, rheumatoid arthritis, and acute pain [3]. Nowadays, CEL is widely used in the treatment of pain, arthritis, and cancers. However, its absorption through oral administration remains a challenge because of its hydrophobicity (log P = 3.5) and very low water solubility (3–7 µg/mL) [4].

Reducing the crystallinity of the drug through formation of solid dispersions to manipulate the solid state of drug is used to improve poor water-soluble compounds’ bioavailability [5]. A eutectic mixture (EM) is a homogeneous mixture of two or more components that usually do not interact to form a new chemical compound but, at certain ratios, inhibit the crystallization process of one another, resulting in a system having a lower melting point than either of the constituents [6,7]. A decreasing melting point means less energy is required for phase change, therefore it becomes easier for EM to dissolve. EM can be formed between Active Pharmaceutical Ingredients (APIs), between APIs and excipients or between excipients, thereby providing a lot of beneficial applications for the pharmaceutical industry [7]. EM, which is formed between API and co-formers, can be considered as a crystalline solid dispersion of a drug in crystalline form carriers [7,8]. For decades, therapeutic eutectic systems for the enhancement of drug bioavailability have emerged as a solution to overcome the drawbacks of some of the existing APIs.

In our previous study, EMs of CEL with adipic acid (ADI) and saccharin (SAC), were identified. A comparison of the disk intrinsic dissolution rate and powder dissolution properties demonstrated that CEL EM significantly increased the dissolution rate compared with CEL and physical mixtures. CEL EM was prepared in bulk by the evaporation crystallization method [6]. Solid CEL EM prepared by the conventional evaporation crystallization method is formed after solvent evaporation and normally requires a milling grinding process to obtain a fine powder. Spray drying has been the preferred method in food and pharmaceutical industries [9]; by rapidly introducing the solution of the active ingredient in organic solvent through a hot gas, the spray drying process often results in fine, dry powder. Moreover, the finer particle size of a drug usually leads to a faster dissolution rate in the body and faster absorption [10]. Nowadays, green and sustainable technologies are the goals of any products and processes. In the pharmaceutical industry, the supercritical CO_2_ solvent anti-solvent (SAS) precipitation method has been regarded as an effective, environmentally friendly, new approach for mass production of fine particles [11]. Hence, it is important to produce CEL EM to enhance CEL bioavailability employing newer, greener techniques such as supercritical fluid (SCF) methods. In this study, we focus on the production of CEL EM using different methods including the employment of supercritical CO_2_. As CO_2_ used with or without the addition of organic solvent presents the definitive advantages of being a “green”, abundant and cheap solvent, it is perfect for application in the production of pharmaceutical products at a temperature near to ambient [11].

Immediate release (IR) dosage forms are types of dosage form that are designed to disintegrate followed by >80% dissolution achieved in 15 min [12]. Superdisintegrants, which provide instant disintegration, are often used in the formulation to improve the performance of the tablets. Quality by design (QbD) implementation in the development of formulations reduces product defects by setting up a quality target product profile (QTPP), process map, risk assessment and control strategy by design of experiment (DoE) [13]. Herein, an attempt was made to develop the design of CEL EM IR tablets.

## 2. Materials and Methods

### 2.1. Materials

Celecoxib (CEL) manufactured by Jiangxi Synergy, China, was kindly gifted from Daehwa Pharm. Co., Ltd. (Seongnam-si, Gangwon-do, Korea).

Adipic acid (ADI) was purchased from Daejung Chemicals & Metals Co., Ltd. (Siheung, Gyeonggi, Korea). Saccharin (SAC) was purchased from Acros Organics (Seoul, Korea).

Ethanol, acetone, methanol, isopropyl alcohol, and chloroform were purchased from Samchun Chemicals (Pyeongtaek-si, Gyeonggi, Korea).

Dimethyl sulfoxide-d_6_ (DMSO-d_6_) was purchased from Cambridge Isotope Laboratories, Inc. (Boston, MA, USA).

Spray-dried lactose monohydrate and Microcrystalline cellulose (Avicel^®^ PH 101) were obtained from Whawon Pharm. Co., Ltd. (Seoul, Korea).

Sodium starch glycolate (SSG) was obtained from Whawon Pharm. Co., Ltd. (Seoul, Korea). Cross-linked carboxy methyl cellulose sodium (cross-linked NaCMC) (Ac-Di-Sol^®^) was purchased from Ashland (Wilmington, DE, USA). Cross-linked polyvinyl N-pyrrolidone (cross-linked PVP) (Kollidon^®^ CL) was purchased from BASF SE (Ludwigshafen, Germany).

Mg stearate was obtained from Whawon Pharm. Co., Ltd. (Seoul, Korea).

Water was purified using Milli-Q^®^ Reference water purification system (Merck Millipore, Alsace, France).

### 2.2. Methods

#### 2.2.1. Screening Assisted Solvent for Eutectic Mixture (EM) Preparation

Acetone, ethanol, methanol, isopropyl alcohol, and chloroform were selected as studied solvents.

At room temperature, 1 g equimolar amounts of CEL and ADI (CEL and SAC) were mixed with each organic solvent (ethanol, methanol, isopropyl alcohol, acetone, chloroform) until totally dissolved. CEL and co-former were dissolved in the minimum amount of solvent (the concentrations ranged from 500 mg/mL to 100 mg/mL based on the solubility of components in the studied solvents) to obtain clear viscous solution. Before any further processing, the dissolved mixtures (CEL-ADI and CEL-SAC) were analyzed by 1H nuclear magnetic resonance (NMR) spectroscopy and compared with the pure compounds.

CEL-ADI and CEL-SAC EM were prepared by using the evaporation crystallization method. Briefly, the resulted mixture (CEL-ADI and CEL-SAC dissolved in solvent) was evaporated at 40 °C for approximately 1 h in a rotary evaporator (Eyela N-1110, Tokyo, Japan) until the solvent was removed, and CEL EM was collected and placed in container for 24 h to completely dry. The CEL EM samples were then evaluated by Differential Scanning Calorimetry (DSC).

#### 2.2.2. NMR Analysis

The NMR analysis was performed to confirm that no chiral discrimination, chemical shifting, peak broadening was detected, which means that there was no change in the chemical structure, no formation of new molecules of the CEL-EM systems [6]. Eutectic solution (CEL-ADI and CEL-SAC dissolved in solvent) and CEL-EM samples in deuterated Dimethyl sulfoxide-d_6_ (DMSO-d_6_) were recorded at 25 °C using JNM-ECZ600R 600 MHz spectrometer (Jeol, Tokyo, Japan).

#### 2.2.3. DSC Analysis

The thermal behavior of the samples was identified by DSC technique using a DSC Auto Q2000 (TA instrument, New Castle, DE, USA). The DSC analysis is considered as a confirmatory analysis as in this thermal analysis, the EM samples’ melting points were confirmed. All samples were accurately weighed to 3–5 mg in an aluminum pan, then were sealed in aluminum pans with lids. Samples were scanned from 40 °C to 200 °C at a heating rate of 10 °C/min under dry nitrogen at a constant flow rate of 40 mL/min. An empty pan was used as reference.

#### 2.2.4. CEL EM Preparation by Spray Drying

The dry powder of CEL-ADI and CEL-SAC EM (CEL, ADI and CEL, SAC at specific weight ratio of 52.7:47.3 and 87.6:12.4—the weight ratios were calculated based on the molar ratios of CEL-ADI and CEL-SAC reported to be 0.30:0.70 and 0.77:0.23 accordant with the Tammann’s triangle by Huyn et al. [6]) were produced by spray drying of 50 mL ethanolic solution of CEL and co-former. The spray drying process was performed by SD1000 spray dryer (Eyela, Tokyo, Japan) under the following set conditions: inlet temperature of 90 °C, feeding rate of 3 mL/min, atomization pressure of 10–20 × 10 kPa, blowing rate of 30 m^3^/h. The airflow is tangential to the feed flow. The spray dryer has a cyclone dust collector, and the solid sample was collected separately in a sample collector (drum). The samples were collected and stored at room temperature for further investigations. The schematic of the spray drying process is shown in Figure 1.

#### 2.2.5. CEL EM Preparation by Supercritical Fluid (SCF) Technique

The schematic of SCF apparatus is shown in Figure 2, the apparatus consists of a CO_2_ cylinder and a reactor. In the cylinder, CO_2_ is in the liquid state; after being compressed to supercritical state, CO_2_ was injected into the reaction vessel and the desired pressure was constantly controlled by a pressure regulator. At the end of the experiment, CO_2_ gas was removed at a stable flow rate managed by a back-pressure regulator.

#### Supercritical Anti Solvent (SAS) Crystallization

CEL, ADI and CEL, SAC at specific weight ratio (52.7:47.3 and 87.6:12.4) were dissolved in 20 mL ethanol in the reactor chamber. The SAS process included 3 stages; (1) mixing stage: The CO_2_ was transferred to the reactor, whereas the pressure was maintained at 16 ± 0.5 MPa. After adjusting to the proper pressure (16 ± 0.5 MPa) and temperature (40 ± 5 °C), the pump was shut down. The mixing time was 30 min for the medium saturation in the reactor by supercritical CO_2_. (2) Washing stage: the solvent (ethanol) was reduced by the presence of supercritical CO_2_. Pump was operated to keep CO_2_ constantly flowing in 20 min, and 3 repetitions were performed totaling 60 min. (3) Venting stage: The pressure was reduced to 0 at a rate of 1 MPa/min. The total time required to depressurize (dynamic time) was approximately 15 min. Finally, CEL EM samples were formed and precipitated on the walls and bottom of the vessel. The samples were collected and stored in desiccators.

#### SCF Assisted Mixing Method

With no organic solvent, CEL-ADI and CEL-SAC EM at the exact weight ratio (52.7:47.3 and 87.6:12.4) were prepared in the reactor chamber and the mixture was mixed using supercritical CO_2_. The SCF-assisted mixing process includes 2 stages: (1) Mixing stage: CO_2_ was injected into the reactor chamber at a constant pressure of 16 ± 0.5 MPa. After adjusting to the proper pressure and temperature, the pump was shut down. The mixing (soaking) time was 24 h. (2) Venting stage: The pressure was reduced to 0 at a rate of 1 MPa/min. The total time required to depressurize (dynamic time) was approximately 15 min. Finally, CEL EM samples were formed and precipitated on the walls and bottom of the vessel. The samples were collected and stored in desiccators.

#### 2.2.6. Powder X-ray Diffraction (PXRD) Analysis

PXRD patterns were measured by Rigaku SmartLAB X-ray diffraction system (Rigaku, Tokyo, Japan) in the θ/2θ scan mode with Cu K-α radiation. The sample was loaded in a small disc-like container and its surface was carefully flattened. θ is the angle between the beam and the crystallographic plane. Samples were run in the range of 3 to 60° with 0.02° step size at a rate of 4°/min.

#### 2.2.7. Fourier Transform Infrared Spectroscopy (FT-IR)

Infrared spectra of the samples were recorded using Cary 630 FT-IR spectrometer (Agilent Technologies, Santa Clara, CA, USA) equipped with an attenuated total reflectance (ZnSe crystal) to check if there were any conformation changes in the EM. Each spectrum was scanned in the range of 400–4000 cm^−1^ with a resolution of 8 cm^−1^, and was derived from single average scans collected in the mid-infrared region (2.5 to 50 μm) at a high spectral resolution; a total of 32 scans were obtained.

#### 2.2.8. Particle Size Analysis

Particle size was measured by Helos-Rodos/VIBRI laser diffraction system (Helos/Rodos; Sympatec GmbH, Clausthal-Zellerfeld, Germany). The RODOS dispenser was operated at 4 bars for de-agglomeration, and HELOS laser sensor was set with R2 lens (detecting range of 0.45 µm–87.5 µm). Particle-size distribution typically includes d_10_, d_50_ and d_90_, which represent the percentage of particles below given size (μm). Volume mean diameter (VDM) and SPAN = $\frac{d_{90}-d_{10}}{2\times d_{50}}$ were calculated.

#### 2.2.9. Scanning Electron Microscopy (SEM)

The shape and surface morphology of the CEL EM were determined by scanning electron microscopy (SEM) (JSM-6700F, JEOL, Tokyo, Japan). Briefly, a small amount of powder was sprinkled onto adhesive carbon tape (Ted Pella Inc., Redding, CA, USA), where the excess powder was gently removed by a jet. The samples were then attached to an aluminum stub and was sputter-coated with gold under vacuum. Photographs were taken at 5× magnification with an accelerating voltage of 1–5 kV to reveal the surface characteristics of the particles.

#### 2.2.10. In Vitro Dissolution Study

Dissolution studies were carried out in an automated dissolution tester (DISTEC 2500) using the USP Apparatus 2 (paddle) method. The bath temperature and paddle speed were set at 37 °C and 50 rpm, respectively. The dissolution medium was 900 mL of hydrochloric acid 0.1 M containing 0.2% (*w*/*v*) sodium lauryl sulfate (SLS) to stimulate the gastro-intestinal fluid. A certain weight of samples equivalent to 200 mg CEL was capsulated, put inside a basket (sinker) then put into the vessels to keep the capsules to not float on the surface of the solution. Samples of dissolution medium were withdrawn through a filter at different time points: t = 5, 10, 15, 20, 30, 45, 60, 90, 120 min. The samples were replaced with fresh dissolution medium of same quantity. Samples were assayed for CEL concentration using high-performance liquid chromatography (HPLC). The HPLC assay was performed using an Agilent-HPLC system 1200 infinity series (Agilent Technologies, Waldbronn, Germany), with a C18 column (CAPCELL, 120 Å pore size, 5 mm, 4.6 mm inside diameter × 250 mm; Shishedo, Tokyo, Japan). The mobile phase comprised 30% 0.2% trifluoroacetic acid (TFA) and 70% ACN, pumped at a rate of 1.5 mL/min. The samples were diluted as required before the injection. The amounts of CEL then were calculated based on the standard calibration curve obtained for CEL at the same condition. The dissolution of each sample was determined in triplicate.

#### 2.2.11. CEL Immediate Release (IR) Tablet Formulation Development by a Quality by Design (QbD) Approach

#### Quality by Design (QbD) Approach

CEL-ADI EM was chosen as the model drug. Novel CEL immediate release (IR) tablets were manufactured (lab-scale) (Table 1) from CEL EM via the direct compression process using QbD approach. The QbD approach was initiated with selection of quality target product profile (QTPP) (Table 2). The risk assessment was performed to select and evaluate the critical material attributes (CMAs) and critical quality attributes (CQAs). The risk assessment (Table 3) was based on prior knowledge, screening experiments, and experience and information on dosage form, obtained from published guidance for industry [14]. In the process map, material attributes (MAs) and process parameters (PPs) were listed (Table 1), CMAs have critical effects on CQAs and among various PPs, critical process parameters (CPPs) which also have critical effects on CQAs, were noted (Table 2). For the final product, appearance, hardness, drug dissolution and disintegration, assay, content uniformity were selected as CQAs. CEL EM (CEL as the active pharmaceutical ingredient (API) and co-former), disintegrant, filler, surfactant, and lubricant were selected as candidate MAs that affect CQAs. Mixing and direct tablet compression were selected as candidate PPs that affect CQAs. Design of experiment (DoE) is employed to determine the interaction between MAs and/or PPs on the performance of the final formulation. Here, a mixture design approach was utilized to investigate the influence of disintegrants used (independent variable) on dependent variables (hardness and drug dissolution). Statistical software JMP^®^ 16 (Statistical Analysis Software, SAS Institute Inc., Charlotte, NC, USA) was employed to perform DoE. Independent variables (factors): sodium starch glycolate (SSG) (X1), cross-linked carboxy methyl cellulose sodium (cross-linked NaCMC) (X2) and cross-linked polyvinyl N-pyrrolidone (cross-linked PVP) (X3) and the dependent variables (response): hardness (kp) (Y1) and dissolution (%) (Y2) were studied and analysis of variance (ANOVA) was performed. X1:X2:X3 is the ratio of SSG, PVP and CMC in the formulation. A total of 12 experiments were run and the design space was created.

#### Preparation of CEL IR Tablets

CEL IR tablets were manufactured using the direct compression process, employing CEL-ADI EM and other ingredients (Table 4). CEL EM was synthesized, checked for NMR, DSC and drug content, then mixed with disintegrant in different proportions, i.e., SSG: cross-linked NaCMC: cross-linked PVP at a ratio of 1: 1: 1, filler and surfactant. Then the mixture was mixed with lubricant; the lubrication time was 2 min. The mixture was direct compressed using ZP-10 rotary tablet press (Shanghai Tianhe Pharmaceutical Machinery, Shanghai, China). The tablets were collected and stored at room temperature until further evaluation.

Measurement of tablet friability: the ability of the tablets to resist chipping and surface abrasion was checked by tumbling the batch in FRV 2000 rotating drum (Copley Scientific, Nottingham, UK). Friability ≤ 0.2% weight loss [16] is considered optimum.

Measurement of tablet hardness: TBF 1000 tablet hardness tester (Copley Scientific, Nottingham, UK) was used to determine tablet hardness. The pressure at which a tablet was crushed was recorded.

Measurement of disintegration time: disintegration time of the tablet was estimated by placing the tablet in distilled water maintained at 37 ± 0.5 °C using a USP disintegration test apparatus (Labfine Instruments, Namyangju-si, Gyeonggi, Korea). The disintegration time was limited to within 120 s.

In vitro drug dissolution study: the release rate of CEL from CEL IR tablets was determined using USP Apparatus 2 (paddle) method. The dissolution test was performed using 900 mL of hydrochloric acid 0.1 M, at 37 °C and 50 rpm. A sample of the solution was withdrawn from the dissolution apparatus at determined time points. The samples were replaced with fresh dissolution medium of same quantity. The samples were filtered and analyzed using HPLC.

Content uniformity tests were performed according to the USP procedures in regard to the dose and ratio of drug substance of the formulations over 25 mg and 25%, respectively.

## 3. Results and Discussion

### 3.1. Assisted Solvent for EM Preparation

Solvents have an effect on the process of making EM as solvents have influence on solubility, stability, chemical reactivity/reaction and/or molecular associations [17]. All the studied solvents are common organic solvents and commercially available; they were selected based on their low boiling point which is suitable for the application of SCF processes. The chosen solvents are expected to dissolve well CEL and its co-former(s) and have no effects on the formation of EM. In addition, the solvent should be volatile enough to be vaporized at ambient condition. The particle size is associated viscosity of solution, thus higher viscosity resulting in larger particle [18].

Among the studied solvents, ethanol, methanol and isopropyl alcohol showed the ability to dissolve CEL, ADI creating the concentrations ranging from 500 mg/mL to 100 mg/mL. Ethanol and methanol showed the ability to dissolve CEL, SAC creating the concentrations ranging from 500 mg/mL to 100 mg/mL.

The clearly distinct peaks in DSC results of methanol assisted CEL-ADI EM and CEL-SAC EM and ethanol assisted CEL-ADI EM and CEL-SAC EM were shown in Figure 3 and Figure 4. Ethanol-assisted EM exhibited the expected melting points in agreement with previous study [6]. With methanol as assisted solvent, the clear endothermic peaks showed that EM was well formed between raw materials, but the increase in the concentration of methanol can increase the melting point of EM. The higher concentrations of solute in solvent normally require less energy to bring them together to solid state. When compared to ethanol, methanol intermolecular force is lower; therefore, its boiling point is lower, but ethanol possesses a stronger dispersion force. Methanol as assisted solvent might affect the thermal stability of the solutes, and the ratio of the solutes to form EM. Further studies will be needed to determine the phase diagram of CEL EM with methanol as assisted solvent. 

### 3.2. Characterization of CEL-EM

#### 3.2.1. DSC Analysis

CEL-ADI EM has melting point at 141± 1 °C and CEL-SAC EM has melting point at 162 ± 1 °C regardless of the different methods of making EM. Ethanol was chosen as solvent.

#### 3.2.2. FTIR Analysis

No evidence (shifting and broadening of peaks) for any bond formation were observed in FTIR results, showing no chemical interaction involved between raw materials or during the process of making EM regardless the different methods.

#### 3.2.3. NMR Analysis

No chiral discrimination, chemical shifting, or peak broadening was detected in NMR results; the NMR results support the observations of the FTIR results, with no new bond formation observed in the EM.

Solid-state NMR spectra are broader, and can give a full effects of anisotropic or orientation-dependent interactions; therefore, we suggest this method to further characterize deeper atomic level structure of the resulted solids.

#### 3.2.4. PXRD Analysis

PXRD analysis was performed SAS, SCF assisted mixing and spray drying CEL EM in comparison with raw CEL and co-former(s) PXRD patterns (Figure 5).

Two binary systems, CEL-ADI and CEL-SAC originally showed sharp peaks in PXRD results and represented that the resultants from the bulk evaporation crystallization EM-making method were in crystalline, not amorphous form [6].

The resulted EMs did not have any observable distinct peaks differing from raw materials, meaning that mixtures do not have a new pattern of crystal lattice and bond. CEL, ADI and SAC have numerous distinct peaks, and all peaks are correlated in complexes regardless of different methods of making EM. It clearly appears that compared to the highly crystalline parent materials, the generated particles are less crystalline and more amorphous, and the degree of crystallinity was as follows: spray drying > SCF assisted mixing > SAS. CEL “re-crystallization” temperature was reported to be at least 30 °C, which is lower than its glass transition at 51.8 °C [19]. There was no significant difference between SAS 3 (45 °C), SAS 2 (40 °C) and SAS 1 (35 °C); thus, there were no significant effects of ambient temprature on amorphous CEL.

In agreement with earlier report of PXRD analysis by Huyn et al. [6], in case of CEL-SAC EM, the amount of SAC was much less than CEL; therefore, the peaks of SAC were hidden by the peaks of CEL in almost every method of making EM (evaporation crystallization, SAS and SCF assisted mixing). In the PXRD pattern of spray drying CEL-SAC sample, the significant deviation in the peak intensity can still be recognized but the low intensities of peaks might indicate that: (a) upon spray-drying at high temprature, all particles’ shapes and orientations were changed drastically, lowering the sample quality; or (b) both SCF-assisted mixing and the spray-drying method somehow disrupted intermolecular interactions of CEL and SAC in the mixture, but was not efficient enough to amorphize both parent substances; SAC was partially amorphized but the amount of SAC in the mixture was low (12.4%). An intermediate amorphous state and differential phase behavior of CEL during spray-drying was also reported with different compositions of solvent system used [20]. 

#### 3.2.5. Particle Size and Morphology Analysis

Figure 6 shows SAS, SCF assisted mixing and spray drying CEL EM under optical microscopy. Obtained SAS crystallization CEL-EM were like snow flakes; the cause of this flake shape is the rapid depressurization of CEL in SCF [21]. The same phenomenon happened to SCF assisted mixing where CEL was dissolved in SCF, but the formed particles were not uniform because the samples went from a completely amorphous state at low tempreture to a nearly amorphous state at ambient temprature; during that process, recrystalization occurred. Recent developent of the SCF spray-drying method permits preparation of dry, fine powders from the SAS process [22,23], but in this study, we compare a single-step processed CEL-EM mixture: the evaporation crystalization method, SAS method, SCF assisted mixing method and spray drying. However, EMs resulting from SCF methods do not have powder form (and EMs resulting from the evaporation crystallization method required an extra grinding/sieving/milling step to reach fine powder form; the particle size of spray drying EM was measured to compared to that of raw CEL fine powder (1.74–8.97 µm). Volume mean diameter (VMD) value showed that the average particle size of spray-dried CEL-ADI EM (5.77 µm) and CEL-SAC EM (4.72 µm) were larger than that of raw CEL (3.25 µm). An additional parameter to show the particle size distribution is the SPAN value, spray drying CEL-ADI EM had the SPAN value (0.68) smaller than that of raw CEL (1.05); the SPAN value of spray drying CEL-SAC (1.08) was almost similar to raw CEL.

An SEM image (Figure 7) of raw CEL fine powder was taken; the image correlates with the assessment of the particle size above. Surface topography of prepared binary EMs by different techniques of CEL and ADI and CEL and SAC examined using SEM was shown in Figure 8a–h. Compared to raw CEL, SAS CEL-EM (Figure 8b,f) appeared to be dominantly needle-shaped crystals. Evaporation crystallization CEL EM (Figure 8a,e) and spray-dried CEL EM (Figure 8d,h) showed small needle-shaped particles aggregated, clumped together probably because of heat application. On the other hand, SCF-assisted mixing CEL EM showed particles shaped like thin plates spread throughout the background; each thin plate particle is about the same size as each raw CEL particle—which can be explained because “recrystallization” occurred right after particle expansion forming amorphous form. Without the use of organic solvent, SCF assisted mixing was similar to the rapid expansion of SCF (RESS) [24]; the expansion of SCF leads to solute precipitation. In this study, CEL and co-former was allowed to dissolve in SCF CO_2_ until uniform, then the pressure was reduced slowly. The solubility of CEL in SCF CO_2_ was reported [25,26], showing that SCF CO_2_ has the ability to dissolve a certain amount of CEL to be micronized in near-saturation condition. In RESS, a fast expansion of CEL and co-former mixture occurs early at the nozzle and precipitation happens in the chamber, in the SCF-assisted mixing method, we allowed 24 h residence time of SCF in the chamber, thus allowing the possibilities of particle recrystallization, agglomeration and post-expansion to happen. The obtained mixture was damp after the process, and was left to desorb in dry surroundings. ADI was also reported to have an increase effect on absorption and desorption of CO_2_ [27]. Overall, coalescing and irregular particles were observed in evaporation crystallization, spray drying and SCF-assisted mixing; otherwise, SAS particles tend to be larger but more homogenous.

#### 3.2.6. Dissolution Test

Dissolution profiles of evaporation crystallization, SAS and SCF assisted mixing CEL EM are shown in Figure 9. In dissolution medium simulated gastric conditions, CEL EM dissolved instantaneously in the first 5 min. At subsequent time intervals, % CEL release increased slowly due to the non-sink nature of the dissolution medium. In the case of CEL-ADI EM, the initial dissolution rate was determined as follows: SAS > spray drying > evaporation crystallization > SCF assisted mixing but at the end of 120 min, 85.4%, 87.61%, 83.2% and 92.15% CEL release was observed according to the order. Forming an EM is a strategy to enhance the dissolution of the drug, more specifically, different methods of making EM mixture forming different new structures thus affecting the depletion of the crystal lattice barrier and the formation of interfacial disorders. We also take into consideration that ADI and SAC were hydrophilic coformers and could not inhibit precipitation. Co-former also affected the perfomance of CEL EM; dissolution profiles of CEL-SAC EM are significantly different compared to CEL-ADI EM ones. For CEL-SAC EM, evaporation crystalization, SAS, SCF assisted mixing and spray drying showed similar fast dissolution rate initially, but after 120 min only the spray-drying sample was closest (92.5%) to fully cumulative CEL release. If we consider that the spray-dried CEL-SAC product was not crystalline EM, but was partially amorphized, the amorphous formulation of drug with the carrier that forms an EM provided higher enhanced drug release. Comparing evaporation crystalization formulations with spray drying, SAS- and SCF-assisted mixing formulations, the particle sizes were bigger; therefore, the dissolving process was not facilitated. The pointy shape and flaky form of SAS- and SCF-assisted mixing increase the surface area of particles, but instead of resulting in clumped particles, SAS crystalization decreased the size to nanometer scale and formed uniform particles. 

#### 3.2.7. Development and Formulation of CEL Immediate Release (IR) Tablet by a Quality by Design (QbD) Approach

A risk analysis was performed (Table 3). Particle size, the amount of lubricant, mixing time with lubricant and compression force may affect the drug product quality. A series of excipients were screened for compatibility and a dual filler system was proposed to achieve the right balance of brittle compression and excipient solubility. Spray-dried lactose monohydrate and microcrystalline cellulose are combined together in tablet formulation as filler with particle sizes of around 50 µm. This combination gives a balance of cohesion and adhesion in direct compression where lactose exhibits brittle fracture and cellulose exhibits plastic deformation [28]. Increasing lubricant amount and its mixing time tended to decrease tablet hardness [29] so the lubricant amount of only 0.5% and the lubrication time of 2 min were chosen. Magnesium stearate was selected as lubricant. Although the tablet hardness increases and friability tended to decrease when compression force was high, the disintegration time was shorter, the tableting process parameters were fixed at slide thick 4.5 mm, before press (pre-compression) 7.5 mm and depth of fill (fill-up) 11.5 mm.

Risk assessment based on experience and risk assessment analysis using Preliminary Hazard Analysis were performed during the process development. This pilot-scale risk assessment indicated that it was highly plausible that not only the particle size, but also the composition of disintegrants affected the dissolution and tableting pressure affected tablet hardness, therefore affecting the dissolution.

Design of experiment (DoE) has been widely used for the design of multi-factor experiments. It provides efficient data collection and helps reduce the workload effectively. In this study, a total of 12 experiments according to the custom design model for CEL IR tablet were performed to obtain the precise design space for mixture of the three disintegrants. More runs will be required with the larger number of factors including disintegrants and other components of the formulation for a mixture design model of the CEL IR tablet. All tested tablets exhibit friability ≤ 0.2% weight loss and disintegration time < 120 s. Dissolution tests and hardness measurements results (responses) were recorded and analyzed by JMP^®^ software. The design space of mixture of disintegrants profile and the prediction of hardness and dissolution were shown in Appendix A. Although the mixture design model for CEL IR tablet was not performed, the custom design model showed that the amount of disintegrants (factors) used alone or in combination affecting the hardness (*p* < 0.0001, R^2^ = 0.99571) and dissolution (*p* = 0.0013, R^2^ = 0.97) of CEL IR tablet statistically significant. Further studies can be performed in a real-life scale-up process; note that in any case of composition changes, the whole process must be re-done. In addition, when it comes to preparation of IR tablets, our QbD approach is limited to MAs at lab-scale (Table 1) but generally, in order to scale-up, the change in PPs is very important due to the adaptation to changes in equipment and environment. For example, in current setting (Appendix A), to reach hardness value of 6 and dissolution at 30-min time point of at least 85%, the ratio of SSG, cross-linked NaCMC and cross-linked PVP is 0.63 (10.08% tablet weight): 0.33 (5.28% tablet weight): 0.03 (0.48% tablet weight), respectively. SSG helps the tablet swell quickly and extensively with minimal gelling and is often recommended to be in the range of 2 to 8% of tablet weight [30]. However, cross-linked NaCMC helps draws off water by capillary action due to its fibrous structure, also minimizes gelling effect and normally only accounts for 1 to 6% weight of a tablet [30]. With the recommendation of 0.5–5% of tablet weight, the rate of swelling of PVP is as high as compare to other disintegrants, it can also facilitate deformation, thus quickly dispersing and swelling in water but not gelling even after prolonged exposure [30]. Surprisingly, the amount of PVP is very small compared to the other compositions and was predicted to be 0 to maximize response values (Appendix A). According to the maximized desirability mode, SSG:Ac-Di-Sol^®^:PVP 0.71:0.29:0 is the optimum disintegrant ratio. Therefore, cross-linked PVP will not be considered primary in the disintegrant mixture. In conclusion, the acceptance range for the disintegrant mixture was shown. According to the Dissolution Testing of Immediate Release Solid Oral Dosage Forms Guidance for Industry, testing water-insoluble or sparingly water-soluble drug products can use a surfactant such as sodium lauryl sulfate (SLS). Dissolution media of CEL capsule consist of 1% SLS [31]; in contrast, we reduced the concentration of SLS to 0.2% in in vitro dissolution study media for CEL-EM (Figure 9). It was noted at this point that the control limit was set a bit too tight (dissolution rate at 30-min time point > 85%) as our CEL-ADI EM IR is expected to be at a higher standard compared to the available marketed ones in terms of dissolution rate aspect. 

## 4. Conclusions

In conclusion, screening for proper solvent to make CEL-ADI and CEL-SAC EM was conducted. Therefore, methods of making EM were developed and initial in vitro dissolution test results were observed and contributed to the comparison between different methods of making EM. Further pharmacokinetic and efficacy studies can also be done to compare those methods. Each methods exhibited both opportunities and obstacles to be scaled up for industrial purposes. In addition, a novel CEL IR tablet was formulated using CEL-ADI EM solid dispersion with the help of JMP^®^ by a QbD approach. The choice of disintegrants used in the formulation was statistically proven to significantly affect the performance of tablets; this is a case study which set an example to follow for many further applications of the QbD approach and of statistics in designing new pharma products.

## Figures and Tables

**Figure 1 pharmaceutics-14-01549-f001:**
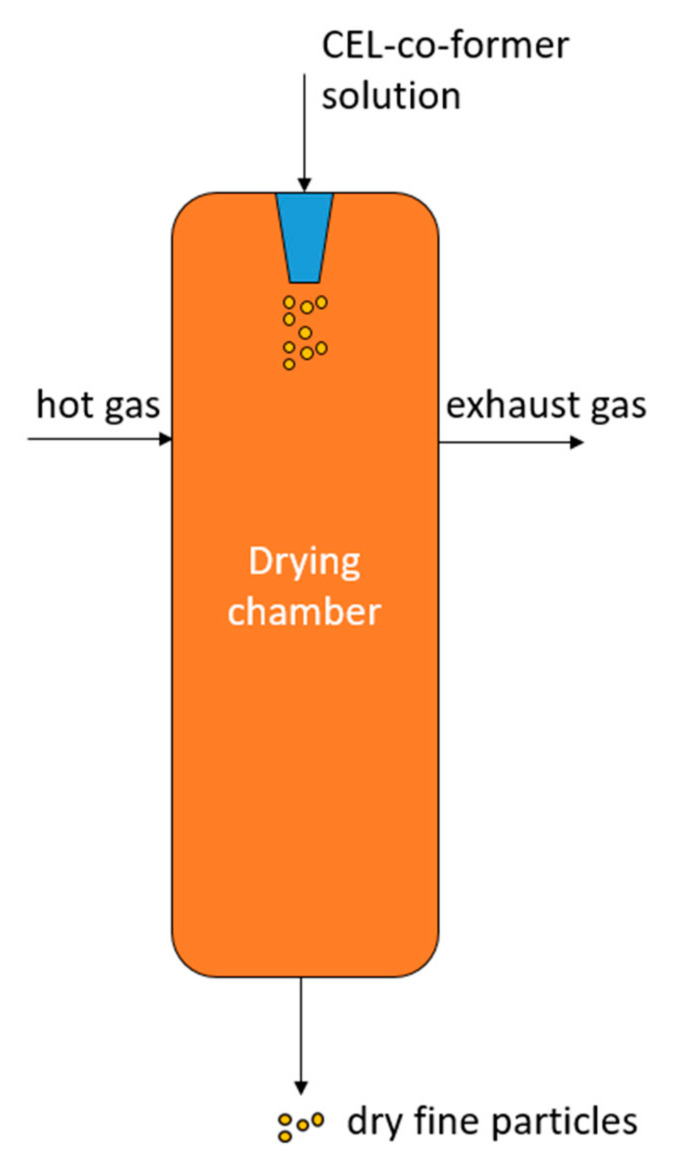
Schematic illustration of the spray drying process.

**Figure 2 pharmaceutics-14-01549-f002:**
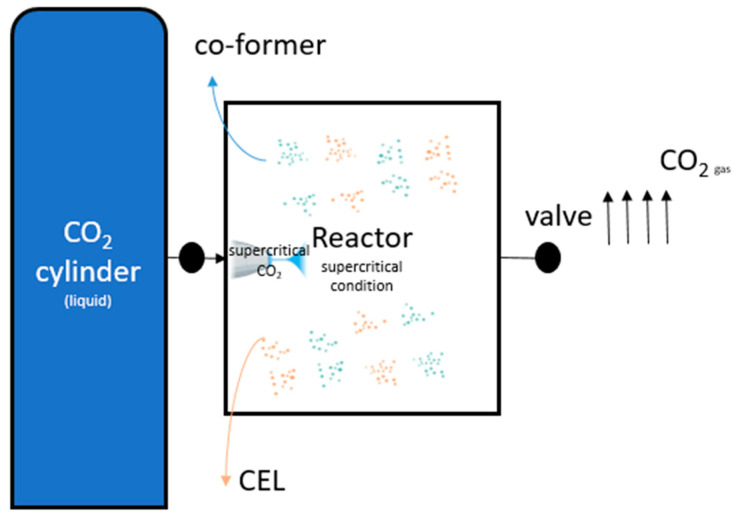
Schematic illustration of the SCF experimental apparatus.

**Figure 3 pharmaceutics-14-01549-f003:**
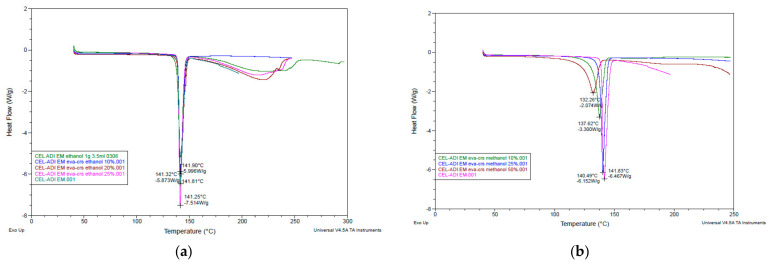
DSC results of CEL-ADI EM: (**a**) ethanol-assisted EM; (**b**) methanol-assisted EM.

**Figure 4 pharmaceutics-14-01549-f004:**
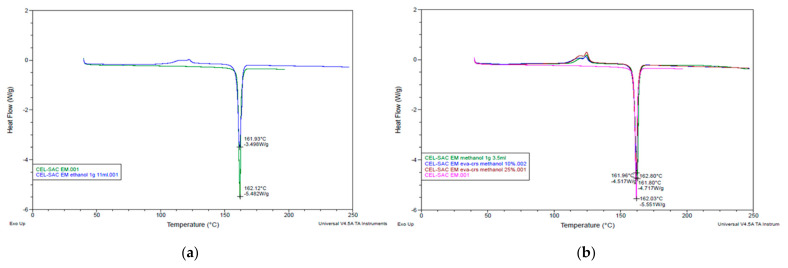
DSC results of CEL-SAC EM: (**a**) ethanol-assisted EM; (**b**) methanol-assisted EM.

**Figure 5 pharmaceutics-14-01549-f005:**
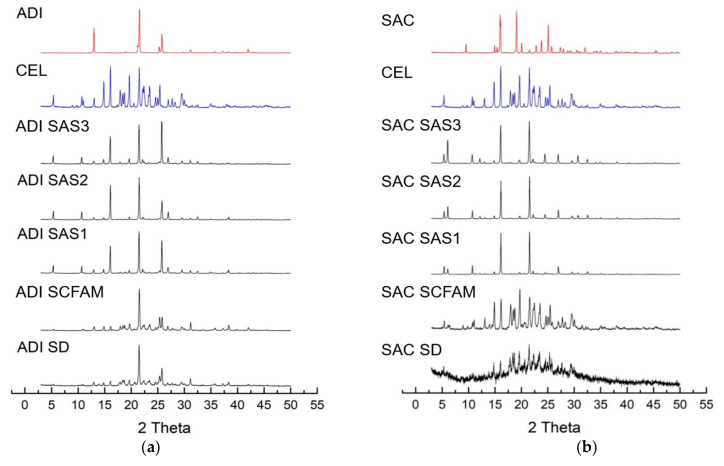
PXRD results of CEL EM: (**a**) CEL-ADI EM; (**b**) CEL-SAC EM. SAS 3 (temperature: 45 °C), SAS 2 (temperature: 40 °C), SAS 1 (temperature: 35 °C), SCFAM: SCF assisted mixing, SD: spray drying.

**Figure 6 pharmaceutics-14-01549-f006:**
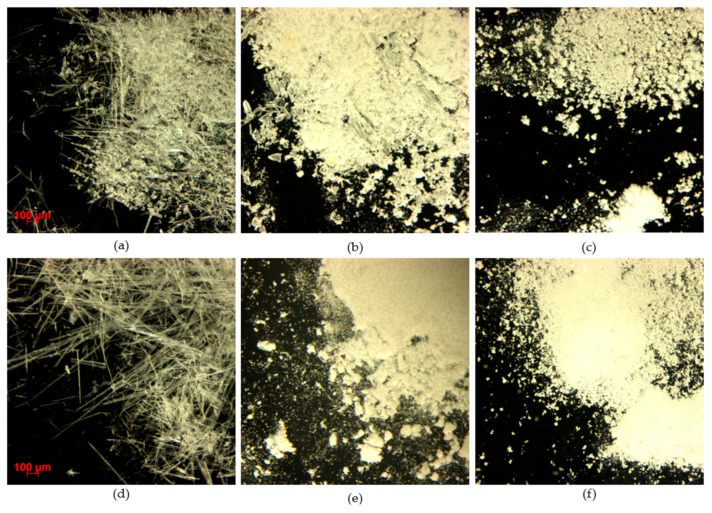
Optical microscopic image of (**a**) SAS CEL-ADI EM, (**b**) SCF-assisted mixing CEL-ADI EM, (**c**) spray-dried CEL-ADI EM, (**d**) SAS CEL-SAC EM, (**e**) SCF-assisted mixing CEL-SAC EM, (**f**) spray-dried CEL-SAC EM.

**Figure 7 pharmaceutics-14-01549-f007:**
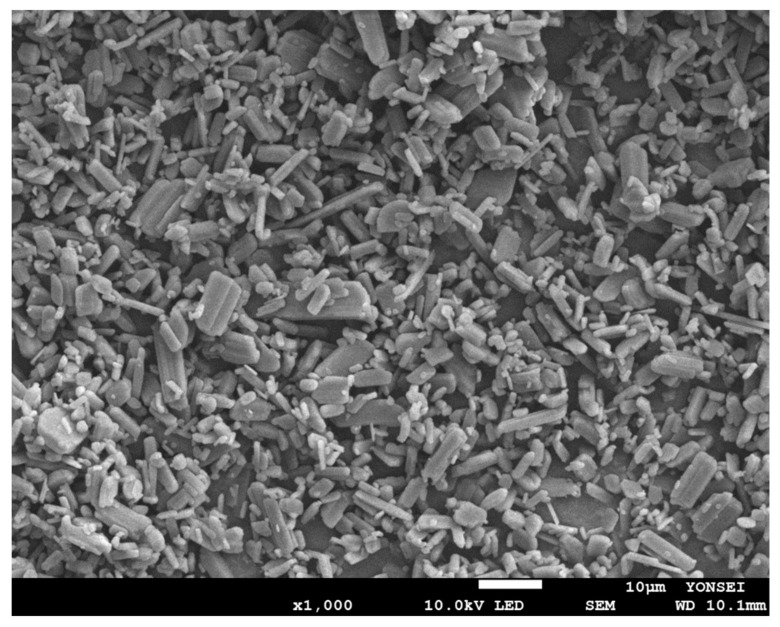
SEM image of raw CEL powder.

**Figure 8 pharmaceutics-14-01549-f008:**
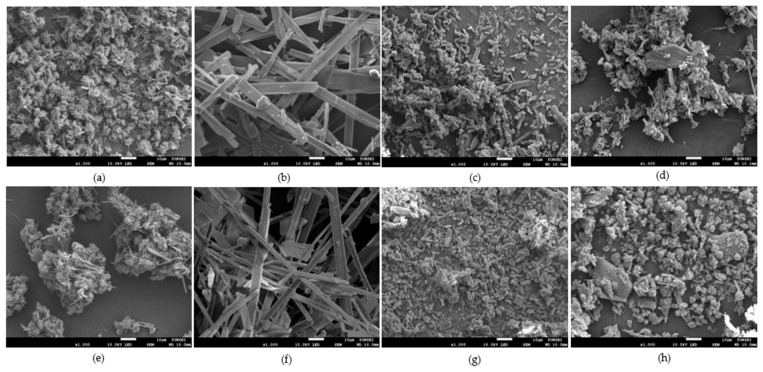
SEM image of (**a**) evaporation crystallization CEL-ADI EM; (**b**) SAS CEL-ADI EM; (**c**) SCF assisted mixing CEL-ADI EM; (**d**) spray drying CEL-ADI EM; (**e**) evaporation crystallization CEL-SAC EM; (**f**) SAS CEL-SAC EM, (**g**) SCF-assisted mixing CEL-SAC EM, (**h**) spray-dried CEL-SAC EM.

**Figure 9 pharmaceutics-14-01549-f009:**
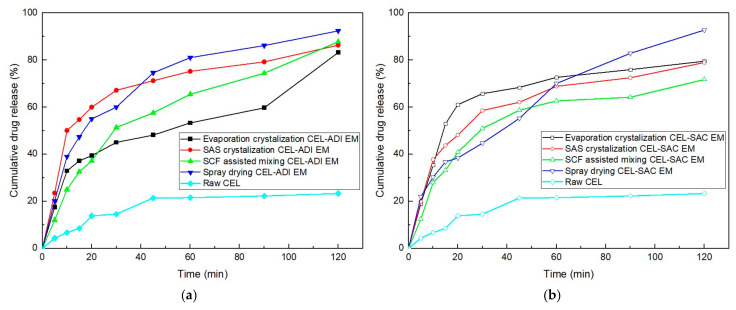
Dissolution profile of CEL EM: (**a**) CEL-ADI EM; (**b**) CEL-SAC EM in comparison with raw CEL dissolution profile. Data were performed as mean (n = 3).

**Table 1 pharmaceutics-14-01549-t001:** Process Map (Lab-scale).

Process	Material	MAs	PPs	Associated CQAs
Mixing	API + co-former (EM) Disintegrants Fillers Surfactant	Particle size Flowability Uniformity Moisture content	Order of addition Hold time Environment (temperature and humidity)	Flowability Uniformity
Mixing	API + co-former (EM) Disintegrants Fillers Surfactant + Lubricant	Particle size Flowability Uniformity Moisture content	Order of addition Hold time Environment (temperature and humidity)	Flowability Compressibility Uniformity Disintegration Dissolution
Direct compresion	API + co-former (EM) Disintegrants Fillers Surfactant + Lubricant	Particle size Compressibility Uniformity	Compressor type Compression force Rotation speed Environment (temperature and humidity)	Appearance (Size and shape) Weight Thickness Hardness Friability Content uniformity Disintegration Dissolution

**Table 2 pharmaceutics-14-01549-t002:** QTPP.

Quality Attributes (QAs)	Target	Critically
Dosage form	Solid oral IR tablet containing 100 mg of CEL (API)	
Appearance	Suitable size and shape	Critical (related to compressibility, thickness, hardness and patient acceptability)
Hardness	6–10 kp	Critical (able to withstand transport, handling, storage)
Friability	≤1% *w*/*w*	Critical (low friability leads to higher hardness of tablets)
Moisture content	≤1%	Not critical, API is not sensitive to hydrolysis
Dissolution	Dissolution acceptance criteria: Q ≥ 80% in 15 min (The International Conference on Harmonization (ICH) Q6A guideline)	Critical (IR tablets enabling T_max_ in less than 2 h)
Disintegration	2.5 to 10 min [15]	Critical (for IR tablets, related to dissolution, disintegration is before dissolution can occur). ICH allows disintegration time with an upper time limit to be used as the drug release acceptance criteria if Q ≥ 80% is achieved in 15 min at pH 1.2, 4.0, and 6.8.
Assay	95–105%	Critical
Content uniformity	Meets USP requirements	Critical

**Table 3 pharmaceutics-14-01549-t003:** Risk assessment.

QAs	Variables			
	API Particle Size	Filler	Disintegrant	Lubricant
Appearance	Low	Low	Low	Low
Content uniformity	Medium	Medium	Low	Medium
Degradation	Low	Low	Low	Low
Disintegration	Medium	Medium	High	High
Dissolution	High	Medium	High	High
Friability	Low	High	Low	Medium
Stability	Low	Medium	Low	Low

**Table 4 pharmaceutics-14-01549-t004:** List of components for CEL immediate release (IR) tablet formulations for DoE mixture design runs. SSG (X1), cross-linked CMC (X2) and cross-linked PVP (X3); independent variables: disintegrant (SSG + PVP + CMC) = 64 mg (16%), respectively.

Ingredient(s) g (%)		Run 1	Run 2	Run 3	Run 4	Run 5	Run 6	Run 7	Run 8	Run 9	Run 10	Run 11	Run 12
API	CEL	100 (25)	100 (25)	100 (25)	100 (25)	100 (25)	100 (25)	100 (25)	100 (25)	100 (25)	100 (25)	100 (25)	100 (25)
co-former	ADI	89.75 (22.4)	89.75 (22.4)	89.75 (22.4)	89.75 (22.4)	89.75 (22.4)	89.75 (22.4)	89.75 (22.4)	89.75 (22.4)	89.75 (22.4)	89.75 (22.4)	89.75 (22.4)	89.75 (22.4)
Filler	lactose monohydrate (75%) & microcrystalline cellulose (25%)	142.65 (35.7)	142.65 (35.7)	142.65 (35.7)	142.65 (35.7)	142.65 (35.7)	142.65 (35.7)	142.65 (35.7)	142.65 (35.7)	142.65 (35.7)	142.65 (35.7)	142.65 (35.7)	142.65 (35.7)
Disintegrant	SSG	32 (8)	-	64 (16)	-	-	-	-	21.3 (5.3)	64 (16)	32 (8)	21.3 (5.3)	32 (8)
Disintegrant	crosslinked NaCMC (Ac-Di-Sol^®^)	-	-	-	-	64 (16)	64 (16)	32 (8)	21.3 (5.3)	-	-	21.3 (5.3)	32 (8)
Disintegrant	crosslinked PVP (Kollidon^®^ CL)	32 (8)	64 (16)	-	64 (16)	-	-	32 (8)	21.3 (5.3)	-	32 (8)	21.3 (5.3)	-
Surfactant	SLS	1.6 (0.4)	1.6 (0.4)	1.6 (0.4)	1.6 (0.4)	1.6 (0.4)	1.6 (0.4)	1.6 (0.4)	1.6 (0.4)	1.6 (0.4)	1.6 (0.4)	1.6 (0.4)	1.6 (0.4)
Lubricant	Mg-stearate	2 (0.5)	2 (0.5)	2 (0.5)	2 (0.5)	2 (0.5)	2 (0.5)	2 (0.5)	2 (0.5)	2 (0.5)	2 (0.5)	2 (0.5)	2 (0.5)

## Data Availability

Not applicable.

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
