# Peer review of "Synthesis of Celecoxib-Eutectic Mixture Particles via Supercritical CO2 Process and Celecoxib Immediate Release Tablet Formulation by Quality by Design Approach"

_pharmaceutics, 2022, doi:10.3390/pharmaceutics14081549_

Round 1

Reviewer 1 Report

The current study by Hong et al., is an attempt to develop a therapeutic eutectic mixture for Celecoxib, a NSAID. The work is valuable and contains important findings and I would like to recommend the publication, however after minor revision and addressing the following questions. The issues in the manuscript are listed as below:

I- Abstract:

It should be more organized, starting with small introduction, aim, method and lastly result

II- Introduction:

1. In line 58, the author mentioned their previous study. In addition, they started to illustrate the method of preparing CEL EM. I think this method is better to be transferred to materials and methods section

2. More references were required in introduction.

III- Method:

1-     In section 2.2.4, why these ratios in line 143 were?

2-     In line 184, what do you mean by that mentioned symbol?

3-     In in vitro dissolution studies, authors mentioned using paddle apparatus then they mentioned basked, please specify?

4-     In line 260, no titles were mentioned to explain the method performed.

5-     Content uniformity test was not discussed. The method should be explained in details.  

IV- Results:

In figure 9, it is better to write the condition of the dissolution media

V- Conclusion

1- It is very short and simple. It should be more detailed

V- References

1-     27 references are small number. If it possible to increase the number.  

Author Response

Dear reviewer,

We are grateful for your thoughtful comments, the contribution to the clarity and accuracy of the manuscript is substantial. Below is a point-by-point description of how we have modified the manuscript according to the reviewer’s comments or otherwise answered the reviewer's questions. On almost all points, we have been able to modify the manuscript exactly as suggested by the reviewers. Our responses are highlighted in Blue and the corresponding revisions in the body of the manuscript are highlighted in Red.

Reviewer 2 Report

The article entitled "Synthesis of celecoxib-eutectic mixture particles via supercritical CO2 process and celecoxib immediate release tablet formulation by Quality by Design approach" is interesting in terms of content and is well written.

To me, there are four important aspects to adress : 

- The main aim of the article is to obtain immediate-release tablet formulation ; however, based on the dissolution rates (figure 9), none of the formulation responded to this aim (e.g. as indicated by the authors : " >80% dissolution achieved in 15 min. The authors should comment on this 

- line 488 : "the CEL-ADI EM IR is expected to be better than the available marketed ones". What are the results obtained for the marketed products?

- line 278 : in vitro dissolution : is the method compondial? or is there a reference to assess the suitability of the method?

- are the conditions of preparation of the proposed formulation transposable to mass production ? please comment on this

Other issues have been detected. Therefore, the authors should read thoroughly their manuscript prior to submitting their revised versions.

For instance :

- line 264 : "was justified"?

- line 450 : "tended" instead of "tented"?

- line 469 : "there were no analys to analyst"?

Author Response

(The authors gave the same response as above.)

Reviewer 3 Report

The article aims to discuss the quality pattern of IR tablets made of eutectic particles of celecoxib. The manuscript contains interesting results that may be useful to improve the drug bioavailability. however, some majors should be addressed before considering for publication. 

1) The authors should clarify if Figure 1 is representative of the instrument used or if it should be considered a very simplified representation. in particular, it is not clear the set-up of the spray drier (e.g., is the airflow tangential to the feed flow? is there a cyclone dust collector?). Similar considerations are valid for Figure 2.

2) The authors have applied a DoE for investigating the impact of factors on the quality profile of final IR tablets. Factors are mainly focused on composition parameters. No process parameters seem to be considered. Since particles are prepared by different methods (e.g., spray drying/SAS/evaporation crystallization/SCF), how have authors investigated the process impact on the DoE responses? The low significance of results may be due to a more consistent impact of process parameters than composition factors. 

3) The native celecoxib powders should be used as a control in dissolution studies

4) The details of the DoE model should be included in the text (p-value, R^2, lack of fit)

5) Lines 466-469. Since the results of the model are not statistically significant, no information can be derived, and the following text seems speculations. 

Author Response

(The authors gave the same response as above.)

Reviewer 4 Report

In the submitted manuscript authors have addressed the QbD approach to the development of celecoxib-eutectic mixtures for further processing via tableting. 

In the last paragraph of the introduction section immediate release is defined as the release of > 80% of API within 15 minutes. Please clarify this claim in the context of the very fast dissolution.

Was the pure drug treated with the supercritical CO2 and tested for dissolution rate? 

Is the proposed concept transferable to other eutectic mixtures as well? More specifically, what makes a eutectic mixture candidate for supercritical CO2 processing? Are solid dispersions always obtained?

Please justify the addition of the surfactant in the dissolution media.

Risk analysis should be more specific to the tested formulations.

Comparison of the pharmaceutical-technical properties of the tested tablet formulations should be clarified in more detail.

According to the Figure 9, none of the formulations has provided the required amount of celecoxib released upon 30 minutes. How can this be explained, in the context of the subsequent discussion on the final tablet formulations.

Author Response

(The authors gave the same response as above.)

Round 2

Reviewer 2 Report

The answers and changes proposed by the authors are appropriate. I recommand this article for publication.

Author Response

Dear Reviewer,
Enclosed please find the revised manuscript entitled “Synthesis of celecoxib-eutectic mixture particles via supercritical CO2 process and celecoxib immediate-release tablet formulation by Quality by Design approach.” for consideration as an article in Pharmaceutics. 
We are grateful for your thoughtful comments. Below is our answers to your questions. On almost all points, we have been able to modify the manuscript exactly as suggested. Our responses are highlighted in Blue in this letter.

Yours sincerely,
Sung-Joo Hwang

Reviewer 3 Report

The authors have improved the manuscript based on the reviewer comments. however, it is reviewer opinion that the dissolution profile of drug substance should be included as control in the dissolution studies, even if the scope of the manuscript is the optimization of the formulation composition. It is true that dissolution profile of drug substance was investigated in ref. 6; however, ref. 6 set up of dissolution study is different from the manuscript. The results obtained cannot be compared with the latter ones. In this light, the Figure 9 should be properly reviewed adding dissolution profile of drug substance as control.

Moreover, The DoE applied by authors is a custom-made and not a mixture design. It seems not to have the same potency of a full mixture design. therefore, the mixture profiler reported in Figure 10 should be deleted; it is misleading.

Author Response

(The authors gave the same response as above.)

Reviewer 4 Report

Authors have addressed the majority of the comments. However, I am still concerned with one of the major claims that has been made - (very) fast CEL dissolution. Please demonstrate the comparison between the commercial product and the optimal formulation dissolution profiles. Otherwise, clarify why the set specifications on the amount of CEL released upon 30 minutes was not met.

Author Response

(The authors gave the same response as above.)

Round 3

Reviewer 3 Report

The authors have improved the manuscript based on the reviewer comments, performing also an additional dissolution experiment with raw drug substances. However, its dissolution profiles was not included in Figure 9. 

The authors should review the Figure including the dissolution profile of raw drug substance as control. 

Author Response

Dear Editors,

Enclosed please find our responses for your comments about our manuscript entitledSynthesis of celecoxib-eutectic mixture particles via supercritical CO2 process and celecoxib immediate-release tablet formulation by Quality by Design approach.” for consideration as an article in Pharmaceutics.

We are grateful for the thoughtful comments which contribute to the clarity and accuracy of the manuscript is substantial. On almost all points, we have been able to modify the manuscript exactly as suggested by the reviewers.

Our responses are highlighted in Blue in this letter.

Yours sincerely,

Reviewer 4 Report

The manuscript can be accepted in its current form.

Author Response

Dear Reviewer,

Enclosed please find our responses for your comments about our manuscript entitledSynthesis of celecoxib-eutectic mixture particles via supercritical CO2 process and celecoxib immediate-release tablet formulation by Quality by Design approach.” for consideration as an article in Pharmaceutics.

We are grateful for the thoughtful comments which contribute to the clarity and accuracy of the manuscript is substantial. On almost all points, we have been able to modify the manuscript exactly as suggested by the reviewers.

Our responses are highlighted in Blue in this letter.

Yours sincerely,
